# An Efficient Multi-center Abdominal Organ Segmentation Network

Chong Wang, Rongjun Ge

College of Computer Science and Technology,
Nanjing University of Aeronautics and Astronautics,Nanjiing,China
`wangchong9905@163.com`

**Abstract.** Efficient abdominal organ segmentation has important clinical significance,but the challenges in label complexity and label professional is extremely limiting this task. Semi-supervised learning (SSL) has impressively improved the label efficiency on medical image segmentation tasks with numerous unlabeled images, but in multi-center situation, its extremely few labeled data enlarges the instability of these methods. In this paper, we designed a new network structure to address multi-organ segmentation from abdominal multi-center and multi-disease CT examinations. To deal with clinical requirements and obtain a low GPU memory but still efficient deep learning model, we use UNeXt which is a Convolutional multilayer perceptron (MLP) based network for image segmentation to achieve abdominal multi-organ segmentation. The network uses tokenized MLPs in latent space reduces the number of parameters and computational complexity while being able to result in a better representation to help segmentation. To ensure the stability of semi-supervised learning in multi-center data, we use an advanced self-training framework(namely ST++). The framework selective retraining via prioritizes reliable unlabeled images based on holistic prediction-level stability. The network structure we use combined with advanced self-training strategy can solve the problem of multi-center data instability in semi-supervised learning and achieve good segmentation effect at the same time. Our experiments show that our method gives strong results on the Dice similarity coefficient, especially for liver and kidney segmentation, and does not require an excessively long inference time.

**Keywords:** Semi-supervised · Multi-center · Abdominal Organ.

## 1 Introduction

In recent years, segmentation has become a crucial task in abdominal image analysis with many applications such as computer-assisted diagnosis, surgery planning. In particular, the precise delineation from Computed Tomography (CT) images of abdominal solid visceral organs including liver, kidneys, spleen and pancreas for localization, volume assessment or follow-up purposes has critical importance. However, the analysis of abdominal imaging datasets is challenging for clinicians since the abdomen is a complex body space. Building a large

labeled abdominal data set requires the expertise of a physician and is time consuming. Therefore, when only few labels are available, the generalization ability of deep learning models are extremely limited.

In this area, semi-supervised learning (SSL) has impressively improved the label efficiency on medical image segmentation tasks with numerous unlabeled images, but in multi-center situation, its extremely few labeled enlarges the instability of these method. Some pseudo-label based methods generate pseudo-labels on unlabeled data sets and improve the quality of pseudo-label generation through deep learning model. However, in multi-center situation, the deep learning model generates unreliable pseudo-labels, which in turn misleads the deep learning model to generate even more unreliable pseudo-labels, resulting in the collapse of learning. Therefore, training a robust computational model with a small amount of labeled data and multi-center unlabeled data is under-investigated.

Few challenges including 2022 FLARE Challenge: Fast and Low-resource semi-supervised Abdominal organ segmentation challenge has been proposed to drive further research on abdominal image segmentation. There are two main objectives in this challenge: the first is to develop a semi-supervised segmentation algorithm that can segment thirteen abdominal organs (liver, spleen, pancreas, right kidney, left kidney, stomach, gallbladder, esophagus, aorta, inferior vena cava, right adrenal gland, left adrenal gland, and duodenum) simultaneously using large amounts of multi-center unlabeled data. The second objective for the participant is to develop an algorithm that not only can perform a high accuracy segmentation but also has high efficiency, meaning it requires low GPU memory and takes reasonable time to run prediction.

Participating in this challenge, we designed an efficient abdominal organ segmentation network to achieve accurate and rapid segmentation of multiple abdominal organs, and we adopt advanced self-training framework(namely ST++ [7]) to solve the instability problem of semi-supervised learning in multi-center data. The network can deduce accurate segmentation results in a reasonable time and consume less resources. In addition, the self-training framework is able to perform selective retraining appropriately to address the stability requirements of this challenging multicenter dataset in semi-supervised learning. The main contributions of this paper are summarized as follows:(1) we are the first combination of UNeXt [6] and ST++ [7] in medical image segmentation, and a stable and accurate segmentation network model is trained using a large number of unlabeled multi-center data;(2) we successfully improve the performance on medical image segmentation tasks while having less parameters, high inference speed, and low computational complexity.

## 2    Method

### 2.1    Preprocessing

To address the challenge of speed and memory, we employed a 2D segmentation model. The input which is in the format of CT image will be splitted into

many slices along its 3D dimension. The segmentation results of these slices will be stacked to form the output. Intensity normalization and random 90 degree reversal is used as pre-processing step. In addition, a z-score normalization is applied based on the mean and standard deviation of the intensity values among the whole training dataset.

## 2.2   Proposed Method

UNeXt [6] is a Convolutional multilayer perceptron (MLP) based network for image segmentation.It in an effective way with an early convolutional stage and a MLP stage in the latent stage.The network is an encoder-decoder architecture with two stages:1) Convolutional stage, and a 2) Tokenized MLP stage.

Each convolution block consists of one convolution layer, batch normalization layer, and ReLU activation. We use kernel size $3 \times 3$, step size 1, and padding of 1. The conv blocks in the encoder use a max-pooling layer with pool window 2 $\times$ 2 while the ones in the decoder consist of a bilinear interpolation layer to up-sample the feature maps. We use the replace transpose convolution with bilinear interpolation basically learnable upsample and contribute to more learnable parameters. In the tokenized MLP block, we move features first and project them into the token. For tokenization, we first use kernel sizes of 3 and change the number of channels to E, where E is the embedding dimension and is a hyperparameter. We then pass these tokens to the MLP, where the hidden dimension of the MLP is the hyperparameter H. Next, features are transmitted through the deep wise convolution layer. We then pass the features through another MLP that converts the dimension from H to O. We use the residual connection here and add the original token as the residuals. We then apply layer normalization and pass the output features to the next block.

The input image is passed through an encoder where the first 3 blocks are convolutionan and the next are 2 Tokenized MLP blocks. The decoder has 2 Tokenized MLP blocks followed by 3 convolutional blocks. Then the features of the three convolution layers are combined into a feature to output the prediction results. Each encoder block reduces the feature resolution by 2, each decoder block increases the feature resolution by 2, and also includes skip connections between the encoder and decoder. The number of channels across each block is a hyperparameter denoted as C1 to C5. For the experiments using UNeXt [6] architecture, we follow C1=32, C2=64, C3 =128, C4=160, and C=256. Figure 1 illustrates the applied UNext [6].

ST++ is an advanced self-training framework, it ensure the stability of semi-supervised learning in multi-center data. From reliable pseudo-label sign-in to unreliable pseudo-label, the label free image is gradually used. And different from the general practice of selecting pixels with high confidence, we select reliable images according to the stability of false labels in the first stage of training. Given labeled dataset $D^l$ and unlabeled dataset $D^u$, first select the most reliable unlabeled image and its pseudo label from $D^u$ to $D^{u1}$, and then retrain on $D^u$ and $D^{u1}$ to obtain a student model $S$. At this point we have used the tagged image and some of the more reliable images and false labels. Then, in

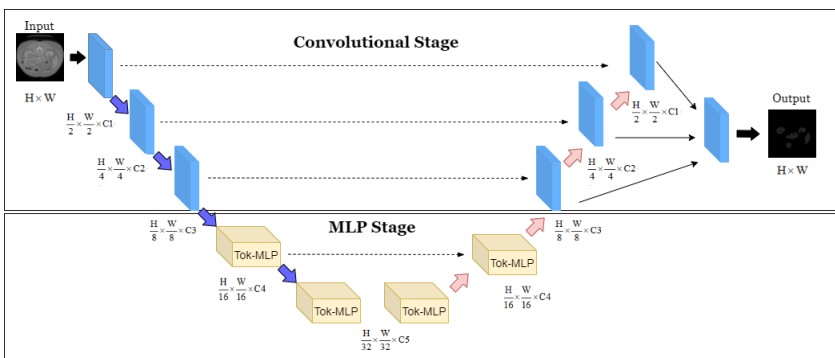

**Fig. 1.** Network architecture

order to make better use of the remaining unreliable images and their pseudo-labels,$D^{u2}=D^u$-$D^{u1}$ we re-label $D^{u2}$ with the learned S.Ultimately retraining on the $D^l$,$D^{u1}$ and $D^{u2}$ results in a final student model for the testing process.

The following describes our selection strategy for reliable tagged images. We observed in the experiment that relatively simple images achieved relatively high accuracy in the early training period, and the pseudo-label change was very small in the late training period. On the contrary, for difficult images, the pseudo-labels predicted by the model in different epochs of training are often quite different. Based on this observation, we propose to determine the reliability of tag-free images and their tagging by measuring the stability of pseudo-labels in different Epochs. To make this measurement strategy more stable, we calculated the meanIOU between pseudo-labels of the whole graph with different epochs. We use the former $k-1$ checkpoint prediction of the $k$ checkpoints and the prediction result of the $k$ checkpoint to calculate meanIOU . The larger the meanIOU is, the more stable the false tag is during training. The quality is also more reliable.We use a combination of binary cross entropy (BCE) and dice loss to train UNeXt [6].The loss $L$ between the prediction $\hat{y}$ and the target $y$ is formulated as:

$$L = 0.5BCE(\hat{y}, y) + Dice(\hat{y}, y)$$

## 3    Experiments

### 3.1    Dataset and evaluation measures

The FLARE2022 dataset is curated from more than 20 medical groups under the license permission, including MSD [5], KiTS [2,3], AbdomenCT-1K [4], and TCIA [1]. The training set includes 50 labelled CT scans with pancreas disease and 2000 unlabelled CT scans with liver, kidney, spleen, or pancreas diseases. The validation set includes 50 CT scans with liver, kidney, spleen, or pancreas diseases. The input which is in the format of CT image will be splitted into

many slices along its 3D dimension.The resolution of all our training images is 512×512.

The evaluation measures consist of two accuracy measures: Dice Similarity Coefficient (DSC), and running effi-ciency measures: running time, area under GPU memory-time curve.

### 3.2   Implementation details

**Environment settings** The development environments and requirements are presented in Table 1.

**Table 1.** Development environments and requirements.

| | |
|---|---|
| Windows/Ubuntu version | Ubuntu 18.04.5 LTS |
| CPU | Intel(R) Xeon (R) CPU E5-2680 @2.40GHz |
| RAM | 16×4GB; 2.67MT/s |
| GPU (number and type) | 1 NVIDIA 3060 12G |
| CUDA version | 11.0 |
| Programming language | Python 3.9 |
| Deep learning framework | Pytorch (Torch 1.7.0, torchvision 0.8.1) |
| Specific dependencies | https://github.com/jeya-maria-jose/UNeXt-pytorch |
| (Optional) Link to code | https://github.com/code-Porunacabeza/flare2022 |

**Training protocols** The training protocols of the baseline method is shown in Table 2. Instead of processing data in a 3D way using 3D patches, our model independently processes 2D axial slices. Image size taken for axial slices is 512×512.

**Table 2.** Training protocols.

| | |
|---|---|
| Network initialization | "he" normal initialization |
| Batch size | 16 |
| Patch size | 512×512 |
| Total epochs | 100 |
| Optimizer | Adam |
| Initial learning rate (lr) | 0.0001 |
| Training time | 28 hours |
| Loss function | BCEDiceLoss |

## 4    Results and discussion

### 4.1    Quantitative results on validation set

As show in Table  3. It can be seen that the use of unlabeled data can improve the segmentation effect.It is worth pointing out that for liver segmentation, the DSC scores has improved 45.25% , indicating great segmentation performance in terms of region overlap between the ground truth and the segmented results. The DSC scores of Gallbladder are 22.00% demonstrating that the small organs and blood vessels contain more segmentation errors, which need further improvement.

**Table 3.** Quantitative results on validation set.

| Organ | DSC(no-unlabeled) | DSC(unlabeled) |
|---|---|---|
| Liver | 0.1425 | 0.5950 |
| Right Kidney | 0.0015 | 0.2654 |
| Spleen | 0.0237 | 0.0893 |
| Aorta | 0.0003 | 0.3967 |
| Gallbladder | 0.2200 | 0.2200 |
| Left Kidney | 0.0109 | 0.3196 |

The GPU usage show in Figure 2.As can be seen from the figure, because we adopted the network model UNeXt [6] with few parameters ,the GPU utilization rate of our proposed model can be controlled below 2GB in inference, and the average inference time is 53 seconds.

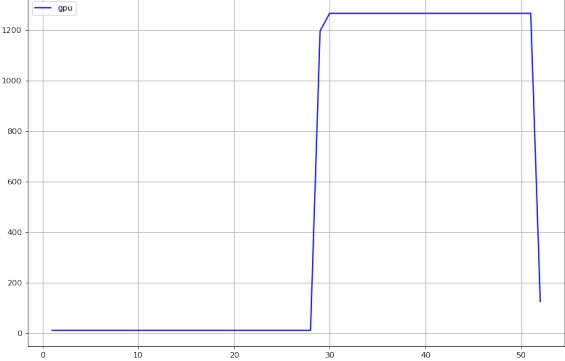

**Fig. 2.** GPU usage

### 4.2   Qualitative results on validation set

Some challenging examples show in Figure 3. It can be found that the method can not segment the small organs well. Some predicted result in the real validation set,(a1) and (a2) are the bad predict performance of our proposed method, (b1) and (b2) are the good predicted results by our proposed method.The method segment the spleen (blue),the right kidney(green) and the liver (red) in this case. As show in Figure 4, the model gets a big boost using unlabeled data.

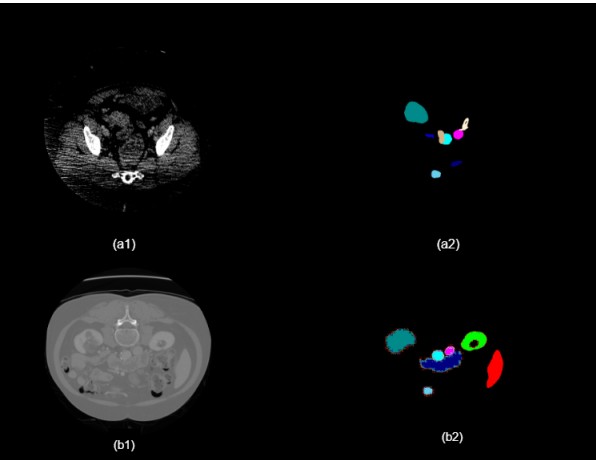

**Fig. 3.** Challenging examples.(a1) and (a2) are the bad predict performance of our proposed method, (b1) and (b2) are the good predicted results by our proposed method.

## 5   Conclusion

The proposed method can work well on cases which is in the same data center as the training set. Besides, the DSC scores of liver segmentation is higher than the other organs, indicating liver maybe a comparable easier task as a result of its bigger size and consistent shape. Disappointing performance is obtained for small organs and blood vessels segmentation as a result of the inter-patient anatomical variability of volume and shape.

The processing of multi-center data is very important. We have tried to use multi-center data to train network in our work. However, the effect achieved is not very obvious, and further analysis is needed. Besides, obtaining an accurate boundary segmentation need further investigate.

**Acknowledgements** The authors of this paper declare that the segmentation method they implemented for participation in the FLARE 2022 challenge has not

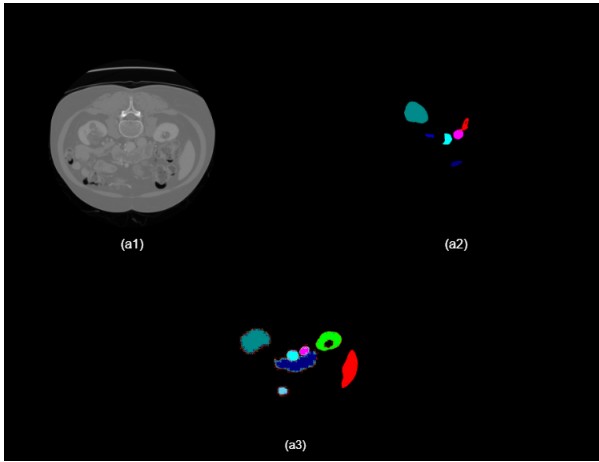

**Fig. 4.** Use unlabeled data and unused labeled data examples.(a1) is the original image,and (a2) is the prediction results for unlabeled images are not used, (a3) is the prediction results for unlabeled images.

used any pre-trained models nor additional datasets other than those provided by the organizers. The proposed solution is fully automatic without any manual intervention.

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
