# OpenReview forum: "An Efficient Multi-center Abdominal Organ Segmentation Network"
_MICCAI.org/2022/Challenge/FLARE_

### Official Review · Reviewer_LqWw · 2022-09-16
**An Efficient Multi-center Abdominal Organ Segmentation Network**

**Rating:** 3
**Confidence:** 2

**Review:**

Strength: The authors proposed a novel semi-supervised pipeline using recently proposed UNeXt along with self-training strategy which is an innovative idea.

Weakness:
1. There so many key elements lost in the paper, for example:
* There are 13 classes to segment, yet the authors only listed 6 of them in the Quantitative Analysis
* In Qualitative Analysis, the ground truth of selected samples are not given
2. Using a 2D network on 3D slices may lead to the loss of dimensional information, the solution or methods to avoid this problem is not given in the paper

---

### Official Review · Reviewer_8pVv · 2022-09-16
**Its mean DSC is not good.**

**Rating:** 3
**Confidence:** 4

**Review:**

In this work, they proposed a UNet like model. however the segmentation is bad especially for spleen(~0.089)
A lot of work can be do in the future!

---

### Meta-Review · Program_Chairs · 2022-09-28

**Recommendation:** Major Revision
**Confidence:** 5

**Metareview:**

Reviewers raise many concerns and suggestions. Please address all comments in the revised manuscript.